# Grading of Scots Pine Seeds by the Seed Coat Color: How to Optimize the Engineering Parameters of the Mobile Optoelectronic Device

**Arthur I. Novikov ***, **Vladimir K. Zolnikov and Tatyana P. Novikova**

Mechanical Department, Voronezh State University of Forestry and Technologies named after G.F. Morozov, 8, Timiryazeva, 394087 Voronezh, Russia; vtis2020@mail.ru (V.K.Z.); novikova_tp.vglta@mail.ru (T.P.N.)
* Correspondence: arthur.novikov@vglta.vrn.ru; Tel.: +7-903-650-84-09

**Abstract:** *Research Highlights*: There is a problem of forest seeds quality assessment and grading afield in minimal costs. The grading quality of each seed coat color class is determined by the degree of its separation with a mobile optoelectronic grader. *Background and Objectives*: Traditionally, pine seeds are graded in size, but this can lead to a loss of genetic diversity. Seed coat color is individual for each forest seed and is caused to a low error in identifying the genetic features of seedling obtained from it. The principle on which the mobile optoelectronic grader operates is based on the optical signal detection reflected from the single seed. The grader can operate in scientific (spectral band analysis) mode and production (spectral feature grading) mode. When operating in production mode, it is important to determine the optimal engineering parameters of the grader that provide the maximum value of the separation degree of seed-color classes. For this purpose, a run of experiments was conducted on the forest seeds separation using a mobile optoelectronic grader and regression models of the output from factors were obtained. *Materials and Methods*: Scots pine (*Pinus sylvestris* L.) seed samples were obtained from cones of the 2019 harvest collected in a natural stand. The study is based on the Design of Experiments theory (DOE) using the Microsoft Excel platform. In each of three replications of each run from the experiment matrix, a mixture of 100 seeds of light, dark and light-dark fraction (*n* = 300) was used. *Results*: Interpretation of the obtained regression model of seed separation in the visible wavelength range (650–715 nm) shows that the maximum influence on the output—separation degree—is exerted by the angle of incidence of the detecting optical beam. Next in terms of the influence power on the output are paired interactions: combinations of the wavelength with the angle of incidence and the wavelength with the grader's seed pipe height. The minimum effect on the output is the wavelength of the detecting optical beam. *Conclusions*: The use of a mobile optoelectronic grader will eliminate the cost of transporting seeds to and from forest seed centers. To achieve a value of 0.97–1.0 separation degree of Scots pine seeds colored fractions, it is necessary to provide the following optimal engineering parameters of the mobile optoelectronic grader: the wavelength of optical radiation is 700 nm, the angle of incidence of the detecting optical beam is 45° and the grader's seed pipe height is 0.2 m.

**Keywords:** seed grading; small-size forest seeds; *Pinus sylvestris* L.; mobile optoelectronic grader; design of experiment; optimal engineering parameters

## 1. Introduction

Normally, grading of Scots pine seeds is performed by size and density. Grading by size is done using a seed sizer with shaker screens. Grading by density is done using an aspirator. When grading by size, the loss of genetic diversity is possible [1]. Accurate seeds' rapid analysis and grading are important for all kinds of mechanized direct seeding [2]—both ground-based [3–6] using tractors and aerial-based [7–11] using unmanned aerial vehicles (UAVs) [12–15]. Automated seed grading [16,17] should reduce the cost of forest landscape restoration [18–22] and ensure the following:

- non-destructive effect on each single seed [23];
- extremely small detecting time for a single seed [24];
- accurate grading based on spectra for each single seed [25,26].

The combined use of optical and microprocessor technology [27] allowed us to develop, with the participation of the authors, design for the mobile optoelectronic grader [28]. When designing it, the above three conditions were taken into account. The principle on which the mobile optoelectronic grader operates is based on the optical signal detection reflected from the single seed. The grader can operate in scientific (seeds spectral band analysis) mode and grading (seeds spectral features grading) mode. When operating in grading mode, it is important to determine its optimal engineering parameters that provide the maximum value of the separation degree into seed coat color classes. For this purpose, a run of experiments was conducted on the Scots pine seeds grading.

## 2. Materials and Methods

### 2.1. Seed Samples

In this study, Scots pine (*Pinus sylvestris* L.) seeds were used. The cones were collected in a Pavlovsk natural stand (Latitude 50.462169; Longitude 40.096446, Altitude 83 m a.s.l.; Voronezh region, Russia) in the fall of 2019. According to the standard seed-processing protocol [29], seeds after extraction from the cones were de-winged in a drum-type wet de-winger (Dewinger 800—BCC AB, Landskrona, Sweden), and then dried in a chamber (DL1200—BCC AB, Landskrona, Sweden) on a moisture level for storage (4.5–7.5 %). Empty seeds were eliminated by gravity method [30] using gravity separator (Mini-Series—BCC AB, Landskrona, Sweden). From the resulting seedlot placed in a glass container with a tightly lapped lid and stored at a temperature of 0 + 5 °C. Immediately before the experiment, the seeds were removed from storage, transported to the laboratory and kept for 24 h at a temperature of 20 °C and a humidity of 40%. Next, the seedlot with a total mass of 1500 g was separated using a stationary photoseparator into Light (L), Dark (D) and Light-Dark (LD) seed-color classes [22]. Between these three classes, the absorption rate in the 650–715 nm wavelength range significantly differed [31]. The seedlot characteristics are shown in Table 1.

**Table 1.** Some physical and physiological features of Scots pine seedlot's color fraction used in this study.

| Parameter | Seed-Color Fraction | | |
|---|---|---|---|
| | **Light (L)** | **Light-Dark (LD)** | **Dark (D)** |
| Munsell's [32] color system [1] | 4.9 YR 7.5/4.2 | 9.8 YR 6.0/4.1 | 7.3 YR 2.6/1.7 |
| CMYKOG's color system [1] | C0, M0, Y35, K26, Or10, G0 | C0, M0, Y14, K40, Or36, G6 | C63, M70, Y85, K54, Or0, G0 |
| Fraction weight, g (% of the seedlot's initial mass) | 361 (24.07) | 1006 (67.07) | 133 (8.86) |
| 1000 seeds weight [2], g at humidity [3] (W, %) | 6.434 (8.36) | 7.869 (7.84) | 7.427 (10.66) |
| Germination [4], % | 96.5 | 96.0 | 94.5 |

[1] These seed classes were classified in the color systems using the digital camera Canon Digital IXUS 100 IS 12.1 MPix (Canon Inc., Tokyo, Japan) for obtaining images and for image processing Digital Color Guide android-software (DIC Corp., Tokyo, Japan). [2] 1000 seeds weight was determined using digital mini scale MTC-series readability 0.001 g (EasyTime Store, Shenzhen, China). [3] Seed moisture content was determined according to the GOvernment STandard of the Russian Federation (GOST RF) number 13056.3-86 [33]. [4] Seed germination was determined according to GOST RF number 13056.6-97 [34].

Preliminary studies of the spectrometric properties of Scots pine seeds in the visible wavelength range [31] indicate a low permeability of the seed outer shell and the need to consider the grading process when detecting radiation reflected from the surface of a single seed [26].

The laboratory unit of a mobile optoelectronic grader for research was built using a three-module scheme [24]. As engineering factors affecting the grading process, the following design and technological parameters of the prototype are determined, independent of

each other: the wavelength of reflected radiation $\lambda_{r(\text{VIS})}$ ($x_1$) [31], the angle of inclination of the detecting beam incident on the seed $\alpha$ ($x_2$) [26] and the height $h_{sp}$ ($x_3$) of the transparent seed pipe [24]. The choice of factors is in good agreement with the results of a priori ranking [35]. For each run of the experiment, three samples of 300 seeds each were selected, including a mixture of randomly selected 100 seeds of each color fraction.

### 2.2. Design of Experiment

Design of Experiment (DOE) is a procedure for selecting the number and conditions of experiments necessary and sufficient to solve the problem with the required accuracy [36]. DOE is also a tool to achieve seed processing cost savings.

The response of the experiment is separation degree of seed-color fraction $\Psi_{cf}$ (formalized $y_1$) described by the equation:

$$\Psi_{cf} = \frac{\Delta n_{cf}}{n_{cf}}, \tag{1}$$

where $\Delta n_{cf}$ is number of graded seeds in the desired seed-color fraction; $n_{cf}$ is number of seeds in the original sample.

The optimum separation of seeds by quality characteristic in the visible wavelength range is equivalent to the mathematical expectation of the degree of separation, tending to 1:

$$Mx[y_1(\text{D, L, LD})] \rightarrow 1. \tag{2}$$

The search for the optimum (1) for the separation of forest seeds, as described in the paper, is effectively approximated by a second-order interpolation polynomial in the region of the extremum of the response function, taking into account linear, quadratic and pair interactions between factors

$$y_1 = b_0 + b_1 x_1 + b_2 x_2 + b_3 x_3 + b_{12} x_1 x_2 + b_{23} x_2 x_3 + b_{13} x_1 x_3 + b_{11} x_1^2 + b_{22} x_2^2 + b_{33} x_3^2. \tag{3}$$

where $\lambda_{r(\text{VIS})}$ ($x_1$) is the wavelength of reflected radiation [31]; $\alpha$ ($x_2$) is the angle of inclination of the detecting beam incident on the seed [26]; $h_{sp}$ ($x_3$) is the height of the transparent seed pipe [24].

According to the DOE theory [36,37] using Microsoft Excel platform, optimization of these grader engineering parameters was performed in the following way:

- choose the type of experiment matrix;
- select levels of factor variation, encode input variables $x_1$, $x_2$, $x_3$ and build a planning matrix;
- complete the planning matrix in coded variables, taking into account quadratic and paired interactions, and supplement it with columns of average response values for each flower-seed fraction;
- calculate the coefficients of the regression equation;
- check the calculated coefficients for significance, first determining the variance of reproducibility, and obtain the regression equation in the encoded variables;
- check the adequacy of the received model;
- interpretation of the resulting model was then performed;
- write out the regression equation in natural variables.

## 3. Results

### 3.1. The Choice of the Tri-Factorial Design, Factors and Levels of Their Variation

A priori, the main factors affecting the output parameters of the separation process in the visible wavelength range are identified: the wavelength of reflected radiation $\lambda_{r(\text{VIS})}$ ($x_1$), the angle of inclination of the detecting beam incident on the seed $\alpha$ ($x_2$) and the height $h_{sp}$ ($x_3$) of the transparent seed pipe [24].

The values of factors and their varying levels are selected (see Table 2).

**Table 2.** Values of factors and levels of their variation in the implementation of a uniform rotatable plan.

| Designation | | Levels of Factor Variation | | | | | Interval |
|---|---|---|---|---|---|---|---|
| **Natural.** | **Coded.** | $-\alpha$ | $-1$ | $0$ | $+1$ | $+\alpha$ | |
| $\lambda_{r(VIS)}$, HM | $x_1$ | 600 | 640 | 700 | 760 | 800 | 60 |
| $\alpha$, degree | $x_2$ | 35 | 39 | 45 | 51 | 55 | 6 |
| $h_{sp}$, m | $x_3$ | 0.10 | 0.14 | 0.20 | 0.26 | 0.30 | 0.06 |
| $x_1 = \frac{\lambda_{r(VIS)}-700}{60}; x_2 = \frac{\alpha-45}{6}; x_3 = \frac{h_{sp}-0.2}{0.06}$ | | | | | | | |

(1) The wavelength of the reflected optical stream in the visible wavelength range $\lambda_{r(VIS)}$.

Preliminary studies [31] of the spectrometric properties of Scots pine seeds using a stationary spectrometer have established that when the wavelength of the detecting radiation incident on the seed is more than 800 nm, as well as less than 600 nm, the spectrum lines of different color-seed fractions are mixed. Therefore, the expediency of choosing the upper and lower limits of factor variation is reduced to the values of 800 and 600 nm, respectively.

(2) The angle of inclination $\alpha$ of the detecting optical beam falling on the seed.

The choice of the experiment center for the factor is based on the theoretical research of the geometric model [26] of optical radiation formation, which established that at $\alpha = 45°$, the optical beam is most effectively reflected from the seed surface and most fully focused into the radiation receiver. Then the factor takes the value 55° at the upper level, and 35° at the lower level.

(3) The height $h_{sp}$ ($x_3$) of the transparent seed pipe.

The center of the experiment for this factor is based on the theoretical research [24] on the movement of a single seed in the mechanical systems of the separation apparatus and is 0.2 m in physical terms. The upper and lower levels are determined based on compliance with the minimum possible overall dimensions of the installation and are 0.3 and 0.1 m, respectively.

*3.2. Implementation of a Tri-Factor Design*

Based on the data in Table 2, we built a planning matrix as Table 3 and implemented it. The planning matrix is based on a full-factor experiment $2^3$, six experiments in the center of the plan, and six experiments at the distance of the "stellar arm" from the center of the experiment. Thus, we consider it quite sufficient to conduct 20 experiments to adequately describe the results of the technological process of grading seeds.

**Table 3.** Planning matrix for testing the efficiency of grading Scots pine seeds using a mobile optoelectronic separator in the visible wavelength range.

| Experience Number | Design Matrix | | | Results of Experience | | | |
|---|---|---|---|---|---|---|---|
| | **Factors** | | | | | | |
| | $x_1$ ($\lambda_{r(VIS)}$) | $x_2$ ($\alpha$) | $x_3$ ($h_{sp}$) | $y_1$ (D) | $y_1$ (L) | $y_1$ (LD) | $\overline{y_1}$ |
| 1 | +1 | +1 | +1 | 0.82 | 0.86 | 0.85 | 0.8433 |
| 2 | −1 | +1 | −1 | 0.8 | 0.83 | 0.87 | 0.8333 |
| 3 | +1 | −1 | −1 | 0.85 | 0.89 | 0.88 | 0.8733 |
| 4 | −1 | −1 | +1 | 0.78 | 0.8 | 0.82 | 0.8000 |
| 5 | +1 | +1 | −1 | 0.81 | 0.85 | 0.83 | 0.8300 |
| 6 | −1 | +1 | +1 | 0.86 | 0.89 | 0.87 | 0.8733 |
| 7 | +1 | −1 | +1 | 0.8 | 0.81 | 0.78 | 0.7967 |

**Table 3.** *Cont.*

| Experience Number | Design Matrix | | | Results of Experience | | | |
|---|---|---|---|---|---|---|---|
| | Factors | | | | | | |
| | $x_1$ ($\lambda_{r(\text{VIS})}$) | $x_2$ ($\alpha$) | $x_3$ ($h_{sp}$) | $y_1$ (D) | $y_1$ (L) | $y_1$ (LD) | $\overline{y_1}$ |
| 8 | −1 | −1 | −1 | 0.79 | 0.78 | 0.75 | 0.7733 |
| 9 | 0 | 0 | 0 | 1.00 | 1.00 | 0.98 | 0.9933 |
| 10 | 0 | 0 | 0 | 0.98 | 1.00 | 0.99 | 0.9900 |
| 11 | 0 | 0 | 0 | 0.99 | 0.99 | 0.99 | 0.9900 |
| 12 | 0 | 0 | 0 | 1.00 | 0.99 | 1.00 | 0.9967 |
| 13 | 0 | 0 | 0 | 1.00 | 1.00 | 1.00 | 1.0000 |
| 14 | 0 | 0 | 0 | 0.98 | 0.99 | 0.99 | 0.9867 |
| 15 | +1.682 | 0 | 0 | 0.95 | 0.98 | 0.97 | 0.9667 |
| 16 | −1.682 | 0 | 0 | 0.94 | 0.96 | 0.98 | 0.9600 |
| 17 | 0 | +1.682 | 0 | 0.89 | 0.87 | 0.88 | 0.8800 |
| 18 | 0 | −1.682 | 0 | 0.88 | 0.89 | 0.87 | 0.8800 |
| 19 | 0 | 0 | +1.682 | 0.93 | 0.97 | 0.92 | 0.9400 |
| 20 | 0 | 0 | −1.682 | 0.94 | 0.95 | 0.91 | 0.9333 |

A planning matrix is constructed taking into account all interactions and average response values as in Table 4.

**Table 4.** Planning matrix with interaction effects for determining the coefficients of the regression equation.

| Experience Number | Factors | | | Interactions | | | | | | Average |
|---|---|---|---|---|---|---|---|---|---|---|
| | $x_1$ | $x_2$ | $x_3$ | $x_1 x_2$ | $x_2 x_3$ | $x_1 x_3$ | $(x_1)^2$ | $(x_2)^2$ | $(x_3)^2$ | $\overline{y_1}$ |
| 1 | +1 | +1 | +1 | 1 | 1 | 1 | 1 | 1 | 1 | 0.8433 |
| 2 | −1 | +1 | −1 | −1 | −1 | 1 | 1 | 1 | 1 | 0.8333 |
| 3 | +1 | −1 | −1 | −1 | 1 | −1 | 1 | 1 | 1 | 0.8733 |
| 4 | −1 | −1 | +1 | 1 | −1 | −1 | 1 | 1 | 1 | 0.8000 |
| 5 | +1 | +1 | −1 | 1 | −1 | −1 | 1 | 1 | 1 | 0.8300 |
| 6 | −1 | +1 | +1 | −1 | 1 | −1 | 1 | 1 | 1 | 0.8733 |
| 7 | +1 | −1 | +1 | −1 | −1 | 1 | 1 | 1 | 1 | 0.7967 |
| 8 | −1 | −1 | −1 | 1 | 1 | 1 | 1 | 1 | 1 | 0.7733 |
| 9 | 0 | 0 | 0 | 0 | 0 | 0 | 0 | 0 | 0 | 0.9933 |
| 10 | 0 | 0 | 0 | 0 | 0 | 0 | 0 | 0 | 0 | 0.9900 |
| 11 | 0 | 0 | 0 | 0 | 0 | 0 | 0 | 0 | 0 | 0.9900 |
| 12 | 0 | 0 | 0 | 0 | 0 | 0 | 0 | 0 | 0 | 0.9967 |
| 13 | 0 | 0 | 0 | 0 | 0 | 0 | 0 | 0 | 0 | 1.0000 |
| 14 | 0 | 0 | 0 | 0 | 0 | 0 | 0 | 0 | 0 | 0.9867 |
| 15 | +1.682 | 0 | 0 | 0 | 0 | 0 | 2.829 | 0 | 0 | 0.9667 |
| 16 | −1.682 | 0 | 0 | 0 | 0 | 0 | 2.829 | 0 | 0 | 0.9600 |
| 17 | 0 | +1.682 | 0 | 0 | 0 | 0 | 0 | 2.829 | 0 | 0.8800 |
| 18 | 0 | −1.682 | 0 | 0 | 0 | 0 | 0 | 2.829 | 0 | 0.8800 |
| 19 | 0 | 0 | +1.682 | 0 | 0 | 0 | 0 | 0 | 2.829 | 0.9400 |
| 20 | 0 | 0 | −1.682 | 0 | 0 | 0 | 0 | 0 | 2.829 | 0.9333 |
| $b_0$ | $b_1$ | $b_2$ | $b_3$ | $b_{12}$ | $b_{23}$ | $b_{13}$ | $b_{11}$ | $b_{22}$ | $b_{33}$ | RE-coefs |
| 0.9958 | 0.005458 | 0.01001 | 0.001065 | −0.0163 | 0.0129 | −0.0163 | −0.03039 | −0.05985 | −0.03982 | |

The coefficients of the regression equation (RE-coefs) are calculated using the corresponding vector columns (see Table 4) and Microsoft Excel. Next, the calculated coefficients are checked for significance, having previously determined the reproducibility variance as the arithmetic mean of the variances of all experiments $S^2_{\{y\}} = 0.000303333$. The mean square deviation of the coefficients of the regression equation is determined:

$$S_{\{y\}} = \sqrt{\frac{S^2_{\{y\}}}{n \cdot m}} = \sqrt{\frac{0.000303333}{20 \cdot 3}} = 0.002248. \tag{4}$$

$$y_1 = 0.9958 + 0.005458x_1 + 0.01001x_2 - 0.0163x_1x_2$$
$$+0.0129x_2x_3 - 0.0163x_1x_3 - 0.03039x_1^2 - 0.05985x_2^2 - 0.03982x_3^2. \tag{5}$$

From the student distribution tables [38] for the number of degrees of freedom $n(m-1) = 40$ at the significance level $\alpha = 0.05$, we found the critical value of the student coefficient $t_{\kappa p} = 2.0211$. Hence, $t_{\kappa p} S_{\{y\}}^2 = 2.0211 \times 0.002248 = 0.004543$. The obtained value was compared with the absolute values of the regression coefficients, as a result of which the significance of all coefficients greater than this value was determined. The exception was the coefficient $b_3$, which is considered insignificant due to the fact that its value is less than the product of the t-test and the variance of reproducibility. In the final form, the regression equation in normalized variables was written as follows:

The adequacy of the obtained model (3) is uniquely described by the ratio of the variance of adequacy $S_{ad}^2$ to the reproducibility variance $S_{\{y\}}^2$,

$$F_{calc} = S_{ad}^2 / S^2\{y\}, \tag{6}$$

called Fischer's calculated F-criterion.

The variance of adequacy was determined by the equation

$$S_{ad}^2 = \frac{n \sum_{j=1}^{N} \left( \overline{y}_j - \hat{y}_j \right)^2}{N - P}. \tag{7}$$

where

$\overline{y}_j$ is the average value of the response from the experiment; $\hat{y}_j$ is the response value calculated from the regression equation;
$n$ is the number of repetitions of each experience from the planning matrix, $n = 3$;
$N$ is the number of experiments according to the planning matrix, $N = 20$;
$P$ is the number of regression coefficients of the analyzed mode, $P = 8$.

From the tables of the *F*-distribution for the significance level $P = 0.05$ and the freedom numbers of the larger variance $f_{ad} = N - P = 20 - 8 = 12$ and the smaller variance $f_y = N(n-1) = 40$, the value $F_{tab} = 2.00$ was found, while the calculated $F_{calc} = 0.49$. In view of the fact that the condition $F_{calc} < F_{tab}$ is met, the regression equation adequately describes the response surface.

The regression model (3) was converted from a normalized representation to a natural one, substituting their expressions using natural variables from Table 3 instead of the factor code

$$y_1 = 0.9958 + 0.005458 \frac{\lambda_{r(VIS)} - 700}{60} + 0.01001 \frac{\alpha - 45}{6}$$
$$-0.0163 \frac{\lambda_{r(VIS)} - 700}{60} \frac{\alpha - 45}{6} + 0.0129 \frac{\alpha - 45}{6} \frac{h_{sp} - 0.2}{0.06}$$
$$-0.0163 \frac{\lambda_{r(VIS)} - 700}{60} \frac{h_{sp} - 0.02}{0.006} - 0.03039 \left( \frac{\lambda_{r(VIS)} - 700}{60} \right)^2$$
$$- 0.05985 \left( \frac{\alpha - 45}{6} \right)^2 - 0.03982 \left( \frac{h_{sp} - 0.2}{0.06} \right)^2 \tag{8}$$

To find the optimal range of values for the desired parameters, we fix the main factors in the center of the plan with free options for the remaining factors. The obtained response surfaces at a fixed value of the optical radiation wavelength (Figure 1), the angle of incidence of the detecting beam (Figure 2) and the height of the seed duct (Figure 3) allow us to note the following.

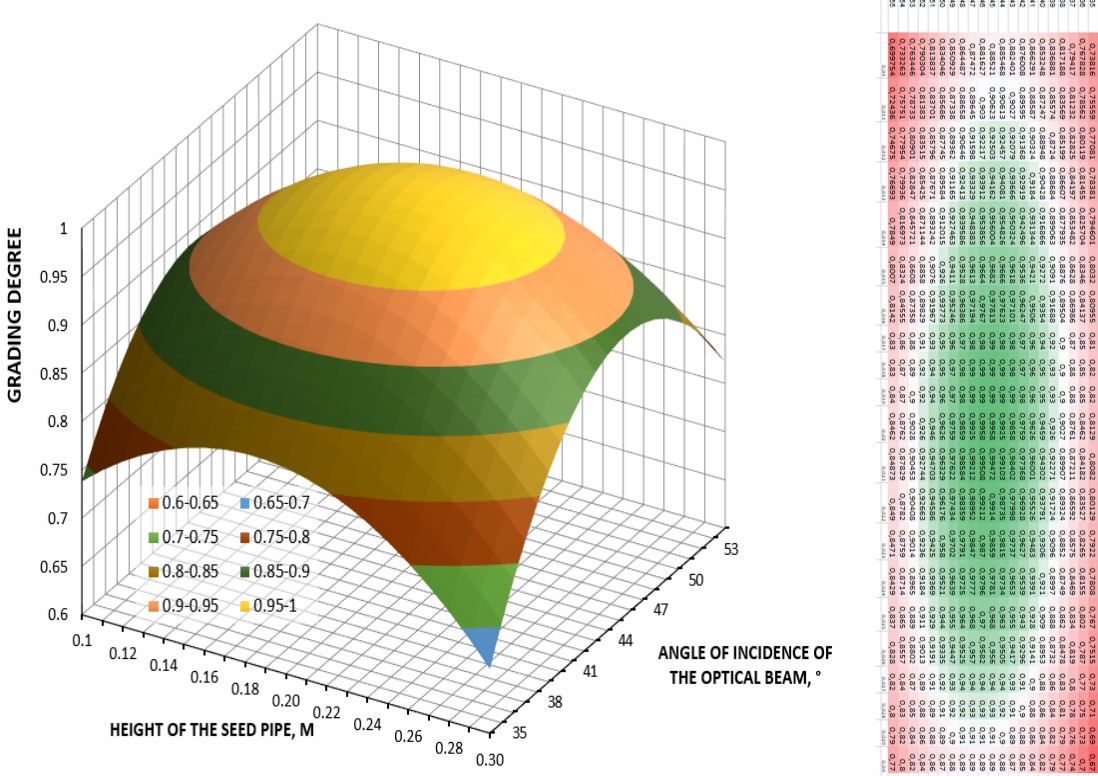

**Figure 1.** The response surface (**left**) and Excel–nomogram (**right**) determine the optimum when implementing a composite full–factor experiment when fixing the wavelength of the reflected radiation in the center of the plan at the level of 700 nm.

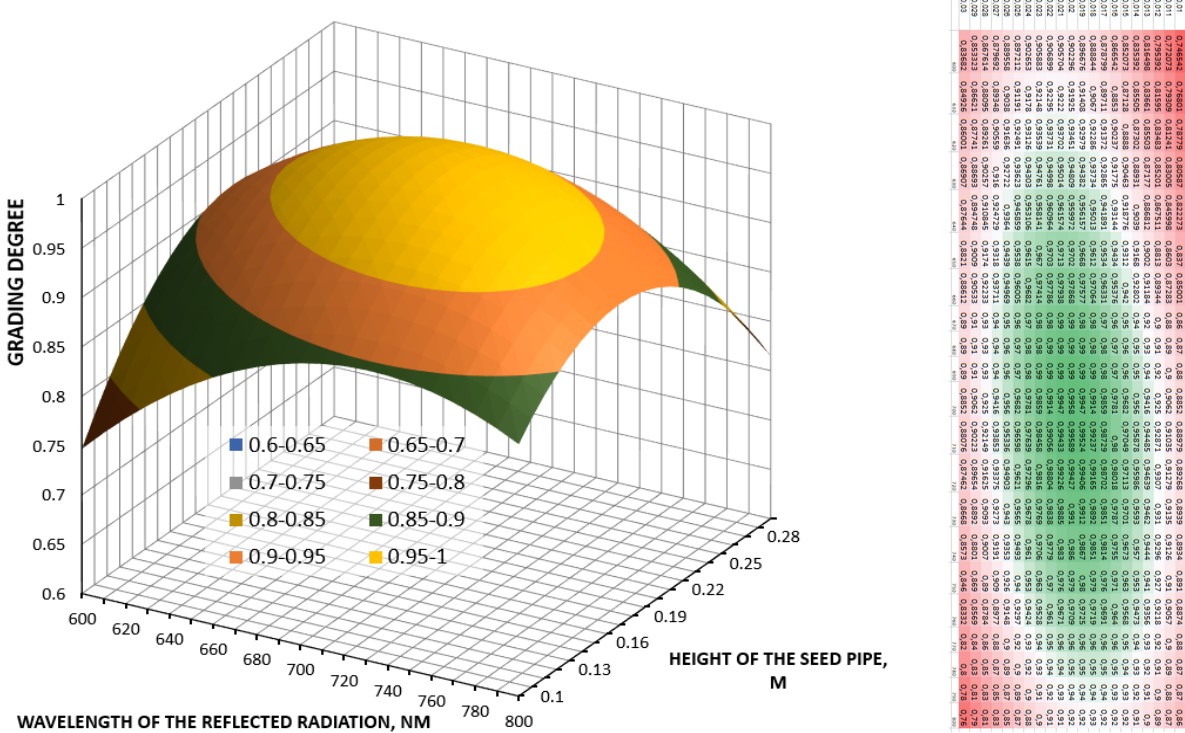

**Figure 2.** The response surface (**left**) and Excel–nomogram (**right**) determine the optimum when implementing a composite full–factor experiment when fixing the angle of incidence of the optical beam in the center of the plan at the level of 45°.

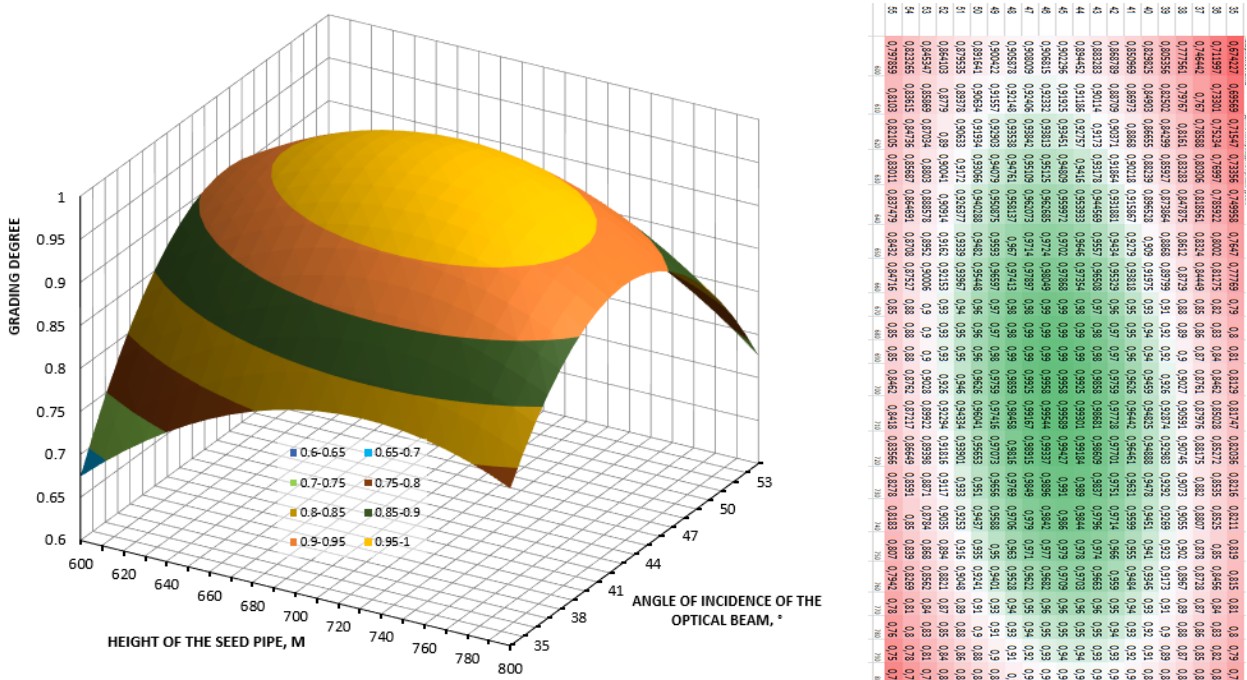

**Figure 3.** The response surface (**left**) and Excel–nomogram (**right**) determine the optimum when implementing a composite full–factor experiment when fixing the height of the seed pipe in the center of the plan at the level of 0.2 m.

## 4. Discussion

A central compositional uniform rotatable plan of the second order was used to plan the full-factor experiment. One of the accuracy characteristics of the regression model is the variance of the $s^2\{y\}$ response values predicted by the regression equation. The rotatability property of the plan means that the accuracy of the regression equation obtained from the results of its implementation is the same at all points in the factor space located at the same distance from the center of the plan. The uniform property in combination with rotatability means that the variance $s^2\{y\}$ is constant in some neighborhood of the center of the plan. Thus, these plans describe the object near the center of the plan with sufficient accuracy.

The structure of the uniform rotatable plan is similar to that of the B-plan [39,40], with the only difference being the number of experiments in the center is set unambiguously (for three factors, it is equal to six) depending on the number of variable factors and based on the requirements of the uniform plan. Another difference from the B-plan is that the values of the factors for the experiments of the orthogonal part of the rotatable plan are located not on the borders of the ranges of their variation, but inside them. Nevertheless, the values of the factors in these experiments are usually denoted $-1$ or $+1$, and the lower and upper levels of each factor are denoted respectively $-\alpha$ and $+\alpha$, where $\alpha$ is a positive number (for three factors $\alpha = 1.682$), called the stellar arm. Thus, each factor in the uniform rotatable plan varies at five levels (see Table 2).

Interpretation of the obtained regression model determines that the maximum influence on the output value of grading quality is exerted by the angle of incidence of the detecting beam. This factor has the largest coefficients for both linear and quadratic variables. Next, in terms of the effect on the response (grading quality), there are paired combinations of the wavelength with the angle of incidence of the detecting beam and the wavelength with the height of the seed pipe, which have coefficients equal in absolute value. The minimum effect on the result is the wavelength of the optical beam. The coefficients of $x_1$, $x_2$, and $x_{23}$ is positive, therefore, with the increase of the values of the factors, but not more than the plan—wavelength visible radiation and angle of incidence of the detecting optical beam, as well as the effect of pair interaction from a combination of

the angle and height of the seed pipe—will increase the degree of separation in the visible range of wavelengths. The coefficients for $x_{12}$, $x_{13}$, $x_{11}$, $x_{22}$, $x_{33}$ have a negative sign, so the response value will increase as these effects decrease.

At a fixed wavelength of 700 nm (see Figure 1), to ensure an effective degree of separation of Scots pine seeds by spectrometric parameters at the level of 0.95–1.0, it is necessary to ensure that the conditions for setting the angle of incident radiation in the range from 44 to 46 degrees with varying heights of the seed pipe from 0.016 to 0.022 m are met.

With a fixed value of the angle of inclination of the optical beam (see Figure 2) falling on the surface of a single seed of 45 degrees to ensure an effective degree of separation of Scots pine seeds by spectrometric parameters at the level of 0.95–1.0, it is necessary to ensure that the conditions for setting the wavelength of optical radiation in the range from 670 to 730 nm with varying heights of the seed pipe from 0.017 to 0.024 m are met.

With a fixed value of the seed pipe height of 0.02 m (see Figure 3), to ensure an effective degree of separation of Scots pine seeds at the level of 0.95–1.0, it is necessary to set the angle of incident radiation in the range from 43 to 47 degrees with a wavelength variation from 680 to 720 nm.

## 5. Conclusions

The optimal values of the engineering parameters of a mobile optoelectronic grader when grading samples of Scots pine seeds will be the following values of variable factors, combinations of which form the extremes of the response surfaces: the wavelength of optical radiation—700 nm; the angle of inclination of the detecting optical beam—45 degrees; the height of the seed pipe—0.2 m.

**Author Contributions:** Conceptualization, A.I.N. and T.P.N.; methodology A.I.N. and V.K.Z.; validation, A.I.N., V.K.Z. and T.P.N.; formal analysis, A.I.N., V.K.Z. and T.P.N.; investigation A.I.N.; data curation, A.I.N., V.K.Z. and T.P.N.; writing—original draft preparation A.I.N., V.K.Z. and T.P.N.; writing—review and editing, A.I.N., V.K.Z. and T.P.N.; visualization, A.I.N. and T.P.N. All authors have read and agreed to the published version of the manuscript.

**Funding:** This research received no external funding.

**Data Availability Statement:** *Pinus sylvestris* L. seeds were used in this study. The cones were collected in a Pavlovsk natural stand (Latitude 50.462169; Longitude 40.096446, Altitude 83 m a.s.l.; Voronezh region, Russia) in the fall of 2019.

**Conflicts of Interest:** The authors declare no conflict of interest.

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
