# Peer review of "Grading of Scots Pine Seeds by the Seed Coat Color: How to Optimize the Engineering Parameters of the Mobile Optoelectronic Device"

_inventions, doi:10.3390/inventions6010007_

Round 1

Reviewer 1 Report

I think the manuscript is of interest for readers. However and in general, it is difficult to read and follow. In my opinion the paper needs to be re-written so that a forest expert like me can understand it. As it is now it is more written as a technical report for involved researchers that already know a lot about the experiments and subject. Thus, major revision before any publication. I have some further remarks that may help improve the paper:

The language needs to be improved by a native English speaking person

The results and discussion should be separated - as it is now it is almost impossible to evaluate the results because they are intermingled with the discussion.

Abstract: L11 - the authors should explain what the device is, and why it is important

Abstract: L18 - No abbreviations in an abstract - thus explain and spell out DOE

L31 - As a reader I wonder how pine seeds normally are graded - there is nothing about this, which is needed in a background/inroduction.

L33 - why only for aereal seeding? It should be important for all kinds of direct seeding. 

L49 -  where the seeds collected in a natural stand? Seed orchard? Provenance? This information is needed if someone wants to replicate the experiment.

L81 - the authors need to explain what DOE is, and spell out the abbreviation the first time it is used.

L87-88 - Models like this must be explained - what are the variables?

L103-227 - Separate results and discussion. And start the discussion try to answer your main question - then discuss the potential drawbacks of the study.

Author Response

The authors sincerely thank the reviewer for his highly professional comments, which significantly contributed to the improvement of the manuscript.

All changes to the manuscript are presented in the PDF-version. To track changes in the MWord-version, please enable the "All corrections" mode on the Review menu.

Response to your comments please see attachment file.

Reviewer 2 Report

Paper is written very well, in adequate scientific and professional levels. However, minor revision is needed. I suggest to make abstract more clear. Abstract must give some results (mainly significantly differences) from the study. Also, it will be nice, if the authors will give some recommendation at the end of abstract. 

Author Response

We thank the reviewer for the valuable comments and provided the suggested additions in the abstract revised version (Please see the attachment).
